# Prognostic Implications of Guideline-Directed Medical Therapy for Heart Failure in Functional Mitral Regurgitation: A Systematic Review and Meta-Analysis [note 1]

**DOI:** 10.3390/diagnostics15050598

**Published:** 2025-03-01

**Authors:** Vasileios Anastasiou, Andreas S. Papazoglou, Stylianos Daios, Dimitrios V. Moysidis, Eirinaios Tsiartas, Matthaios Didagelos, Kyriakos Dimitriadis, Theodoros Karamitsos, George Giannakoulas, Konstantinos Tsioufis, Antonios Ziakas, Vasileios Kamperidis

**Affiliations:** 11st Department of Cardiology, School of Medicine, Faculty of Health Sciences, Aristotle University of Thessaloniki, 54636 Thessaloniki, Greece; vasianas44@gmail.com (V.A.); stylianoschrys.daios@gmail.com (S.D.); manthosdid@yahoo.gr (M.D.); karamits@gmail.com (T.K.); g.giannakoulas@gmail.com (G.G.); tonyziakas@hotmail.com (A.Z.); 2Athens Naval Hospital, 11521 Athens, Greece; anpapazoglou@yahoo.com; 3424 General Military Hospital, 56429 Thessaloniki, Greece; dimoysidis@gmail.com; 4Institute of Clinical Trials and Methodology, Faculty of Population Health Sciences, University College London, London WC1H 4AJ, UK; 5School of Medicine, National and Kapodistrian University of Athens, Hippokration General Hospital, 11527 Athens, Greece; dimitriadiskyr@yahoo.gr (K.D.); ktsioufis@hippocratio.gr (K.T.)

**Keywords:** functional mitral regurgitation, heart failure guideline-directed medical therapy, prognosis, meta-analysis

## Abstract

**Objectives:** Randomized evidence on the role of heart failure guideline-directed medical therapy for patients with functional mitral regurgitation (FMR) is lacking. The present meta-analysis sought to investigate the prognostic impact of different pharmacotherapy categories recommended in heart failure on subjects with FMR. **Methods:** A systematic literature review was conducted to identify studies reporting the association of renin angiotensin system inhibitors (RASi), beta-blockers (BB), and mineralocorticoid receptor antagonists (MRA) with outcomes in FMR. A random-effects meta-analysis was conducted to quantify the unadjusted and adjusted hazard ratios [(a)HRs] for all-cause death and the composite outcome in each medical category. **Results:** Twelve studies with 6,715 FMR patients were included. The use of RASi and BB was associated with a significantly lower risk of all-cause mortality (HR 0.52 [0.39–0.68]; *p* < 0.00001, I^2^ = 62% and HR 0.62 [0.49–0.77]; *p* < 0.0001, I^2^ = 44%, respectively) and the composite outcome (HR 0.54 [0.44–0.67]; *p* < 0.00001, I^2^ = 33% and HR 0.62 [0.52–0.75], *p* < 0.00001, I^2^ = 35%, respectively) in unadjusted models. Both RASi (aHR 0.73 [0.56–0.95], *p* = 0.02, I^2^ = 52%) and BB (aHR 0.60 [0.41–0.88], *p* = 0.009, I^2^ = 55%) retained their association with the composite outcome in pooled adjusted models. The prognostic benefit of using RASi or BB was retained in subgroup analyses including only (1) patients with moderate or severe FMR and (2) patients with reduced or mildly reduced left ventricular ejection fraction. MRA did not demonstrate a significant association with improved outcomes. **Conclusions:** RASi and BB administration appear to have a favorable prognostic impact on patients with FMR, regardless of the severity of regurgitation.

## 1. Introduction

The prevalence of functional mitral regurgitation (FMR) is rising in parallel with the prevalence of heart failure (HF), and has been reported in up to 59% of patients with HF [1]. More specifically, the European Society of Cardiology (ESC) Heart Failure Long-Term Registry showed a prevalence of FMR of up to 35% in the HF with reduced ejection fraction (HFrEF) group, 30% in the HF with mildly reduced EF (HFmrEF) group, and 20% in the HF with preserved EF (HFpEF) group [2]. Other studies have also demonstrated a prevalence of severe FMR of up to 45% in patients with advanced HFrEF [3,4]. Any grade of FMR is independently associated with adverse outcomes in HF and confers a poor prognostic course throughout the range of left ventricular EF (LVEF) [5,6,7].

Therapeutic management of FMR remains ambiguous, with updated recommendations advocating the use of guideline-directed medical therapy for HF (GDMT-HF) as the first step, and introducing mitral valve surgery or transcatheter edge to edge repair for appropriate candidates who remain symptomatic [8]. However, randomized control trials focusing on the role of GDMT-HF in patients with FMR are lacking, and concomitantly the randomized trials that established the use of GDMT-HF systematically excluded the patients with significant valvular heart disease [9,10]. Observational data have shown that GDMT-HF may induce cardiac remodeling and functional recovery leading to a decrease in FMR by restoring the leaflet coaptation [11,12]. Whether this GDMT-HF effect translates into a clinical benefit for patients with FMR remains poorly defined.

Since the available evidence regarding the prognostic role of GDMT-HF for patients with FMR is scarce, the present meta-analysis sought to quantitatively synthesize all available literature regarding the independent long-term prognostic impact of different medical categories recommended for HF on patients with FMR.

## 2. Methods

The current systematic review and meta-analysis was performed in accordance with a prespecified research protocol registered in the PROSPERO database (CRD42024500616, https://www.crd.york.ac.uk/prospero/display_record.php?RecordID=500616, accessed on 22 January 2025). The reporting follows the Preferred Reporting Items for Systematic reviews and Meta-Analyses (PRISMA) 2020 reporting guidelines (Appendix A) [13].

### 2.1. Literature Search

The literature search was conducted in PubMed, Scopus, and Web of Science databases from database inception until 2/2025. Basic keywords used in search strings were (“functional mitral regurgitation” OR “secondary mitral regurgitation” OR “ischemic mitral regurgitation” OR “heart failure”) and (“angiotensin-converting enzyme inhibitor” OR “angiotensin receptor blocker” or “angiotensin receptor-neprilysin inhibitor” OR “b-blocker” OR “mineralocorticoid receptor antagonists” OR “medical therapy” OR “medical management” OR “optimal medical management” OR “OMT” OR “guideline-directed medical therapy” OR “GDMT”) and (“prognosis” OR “outcome”) in both free text and Medical Subject Headings (MeSH) format. The reference lists of the eligible studies and relevant reviews were manually searched to identify any papers not previously detected.

### 2.2. Outcomes of Interest and Treatment Categories

The primary study outcome was the difference in all-cause mortality rates between patients with FMR receiving or not receiving a certain medical category of the GDMT-HF. The secondary study outcome was the observed differences in the rate of composite adverse events including HF-related hospitalization, left ventricular assist device implantation, heart transplantation, mitral valve intervention, and all-cause mortality between patients receiving or not a certain medical category of the GDMT-HF.

GDMT-HF was categorized into three medical categories: (i) renin angiotensin system inhibitors (RASi) [14], including angiotensin-converting enzyme inhibitors, angiotensin receptor blockers, and an angiotensin receptor/neprilysin inhibitor, (ii) b-blockers (BB), and (iii) mineralocorticoid receptor antagonists (MRA) [15].

Each outcome was individually assessed in each medical category of the GDMT-HF when data synthesis could be performed, comparing FMR patients receiving with FMR patients not receiving each medical category.

### 2.3. Eligibility Criteria

Observational (prospective or retrospective) cohort or randomized controlled studies were included in the meta-analysis if they provided information on the association of all-cause mortality and/or on the composite outcome with GDMT-HF for patients with HF and at least mild FMR. No restrictions were applied with respect to HF etiology or left ventricular ejection fraction (LVEF). Exclusion criteria of the meta-analysis included the following: (1) abstracts, case reports, reviews, editorials, and practice guidelines; (2) studies conducted in patients with congenital heart disease; (3) studies not providing outcome data or not reporting unadjusted and/or adjusted hazard ratio [(a)HR] with 95% confidence intervals (CIs) for the outcomes of interest; and (4) studies published exclusively in a non-English language.

### 2.4. Quality Assessment

The methodological quality of the included studies was evaluated using the Quality in Prognosis Studies (QUIPS) tool, as recommended by the Cochrane Collaboration [16]. The risk of bias for each eligible study was evaluated in each of the following domains as “low”, “moderate”, or “high”: study participation, study attrition, prognostic factor measurement, outcome measurement, study confounding, statistical analysis, and reporting. The risk of bias assessment was conducted by two independent reviewers (VA and SD) and any disagreements were solved by consensus.

### 2.5. Data Extraction and Data Synthesis

For all eligible studies, two independent reviewers (VA and ET) extracted information on the study design, study size, source of data, population characteristics, pharmacotherapy, duration of follow-up, outcomes of interest, adjusted and unadjusted HR and CIs for the outcomes of interest, and adjustment for confounding factors.

For the meta-analysis, the respective (a)HRs was used for all-cause mortality and/or the composite outcome of patients with at least mild FMR receiving a medical category of the GDMT-HF against those not receiving it Random effects models were used with the inverse variance method. For both primary and secondary outcomes, the results are presented as pooled (a)HRs and 95% Cis with a two-sided significance level of *p* < 0.05. The heterogeneity of each analysis was quantified by calculating I^2^ and was classified as low (<25%), moderate (25% to 75%), or high (>75%) [17]. The possibility of publication bias was examined by applying the funnel plot method. The Egger’s test was not applicable due to the limited number of eligible studies in each meta-analysis (less than 10) [18].

Furthermore, subgroup analyses were performed to address the risk of all-cause mortality and/or the composite outcome only (1) for patients with significant (moderate or severe) FMR and (2) for patients with LVEF < 50%. A sensitivity analysis excluding studies that reported including patients with multivalvular pathology was conducted. Meta-regression analyses could not be performed because of the limited number of eligible studies in each meta-analysis (less than 10). The Review Manager 5.4 software was used for the statistical analysis and visualization of our findings.

The quality of evidence provided by the analyses was assessed through the Grading of Recommendations Assessment, Development, and Evaluation (GRADE) tool [19]. The quality of evidence in each domain (risk of bias, inconsistency, indirectness, imprecision, probability of publication bias, plausibility of residual confounding, dose–response gradient, and magnitude of effect) was assessed by two independent reviewers (A.S.P. and VA) depending on the existence of “very serious”, “serious” or “not serious” concerns. The final outcome (i.e., certainty and importance of the study outcomes) was automatically generated via the GRADE pro Guideline Development Tool [(Software), McMaster University and Evidence Prime, 2023; available from gradepro.org].

## 3. Results

### 3.1. Search Outcomes

The study selection process is summarized in Figure 1. The screening of databases identified a total of 3721 studies, and of these, 150 were assessed as a full text for eligibility. Most studies were off-topic, whereas others were excluded as non-eligible. The reasons for exclusion included not reporting outcomes of interest, the inability to synthesize reported outcome data, the inclusion of other medication than the pre-specified, or the inclusion of non-FMR cohorts (Appendix A). The final sample comprised 12 studies with 6715 FMR patients [20,21,22,23,24,25,26,27,28,29,30,31].

### 3.2. Study Characteristics

The baseline characteristics of the studies included are summarized in Table 1. Two studies exclusively included patients with ischemic FMR [20,21]. Three studies comprised FMR patients undergoing transcatheter edge to edge repair [23,24,28]. Two had FMR patients with acute decompensated HF [25,26], and the rest mixed FMR cohorts [22,27,29,30,31]. Overall, 8 out of 12 studies included only moderate/severe FMR [21,22,23,24,28,29,30,31], while 4 studies also included patients with mild FMR [20,25,26,27]. Five studies provided outcome data on all three prespecified categories of medical therapy [23,24,25,27,28], three provided data on RASi and BB [20,26,29], three only on BB [22,30,31], and one only on RASi [21]. The quality assessment indicated an overall low risk of bias for six of the included studies [21,23,25,26,29,30], whereas the other six were considered of moderate risk of bias [20,22,24,27,28,31] mainly driven by domains of study attrition, outcome measurement, and study confounding (Appendix A).

### 3.3. Outcome Analyses

#### 3.3.1. Renin–Angiotensin System Inhibitors

Data from nine studies were synthesized to demonstrate the prognostic impact of RASi in FMR [20,21,23,24,25,26,27,28,29]. The use of RASi was associated with a significantly lower risk of all-cause mortality in unadjusted models (pooled HR 0.52 [0.39–0.68]; *p* < 0.00001, I^2^ = 62%) (Figure 2A), but this association did not retain statistical significance after adjustment (pooled aHR 0.73 [0.37–1.44], *p* = 0.36, I^2^ = 82%) (Figure 2B). Patients receiving RASi had a significantly lower risk of composite adverse events both in unadjusted (pooled HR 0.55 [0.44–0.67]; *p* < 0.00001, I^2^ = 33%) (Figure 2C) and in adjusted modes (pooled aHR 0.73 [0.56–0.95], *p* = 0.02, I^2^ = 52%) (Figure 2D). When subgroup analysis was performed, including only studies with moderate or severe FMR, the use of RASi retained a significant association with lower risk both for all-cause mortality (Figure 3A) and composite adverse events (Figure 3B,C). In the subgroup analyses using only studies with LVEF < 50%, the use of RASi was linked with a decreased risk of all-cause mortality risk (in the unadjusted models), and with a decreased risk of composite adverse events (in both unadjusted and adjusted models). Finally, the sensitivity analyses omitting studies that reported to include patients with multivalvular pathology did not yield different outcomes than the main analyses.

#### 3.3.2. Beta-Blockers

Eleven studies provided appropriate outcome data on BB for patients with FMR [20,22,23,24,25,26,27,28,29,30,31]. Patients receiving BB had a 38% lower risk of all-cause mortality (pooled HR 0.62 [0.49–0.77]; *p* < 0.0001, I^2^ = 44%) (Figure 4A) and a 38% lower risk of composite adverse events (pooled HR 0.62 [0.52–0.75], *p* < 0.00001, I^2^ = 35%) (Figure 4B) compared to patients not taking BB. The risk of composite adverse events remained significantly lower for those receiving BB even after adjustments for clinically relevant factors (pooled aHR 0.60 [0.41–0.88], *p* = 0.009, I^2^ = 55%) (Figure 4C). The subgroup analysis for patients with moderate or severe FMR demonstrated similar outcomes, with the use of BB being linked with a significantly lower risk of all-cause mortality and composite adverse events (Figure 5). In the subgroup analyses using only studies with LVEF<50%, the use of BB was linked with a decreased risk of all-cause mortality risk and composite adverse events. The sensitivity analyses excluding studies that reported to have patients with multivalvular pathology did not yield different outcomes to the main analyses.

#### 3.3.3. Mineralocorticoid Receptor Antagonists

Data from five studies could be synthesized to quantitatively assess the role of MRA in FMR [23,24,25,27,28]. There was a trend for a lower risk of all-cause mortality (pooled HR 0.88 [0.75–1.05]; *p* = 0.16, I^2^ = 0%) (Figure 6A) and composite adverse events (pooled HR 0.95 [0.81–1.12]; *p* = 0.55, I^2^ = 18%) (Figure 6B) for FMR patients receiving MRA compared to their counterparts not receiving MRA; however, this association did not reach the significance threshold. Similarly, when the subgroup analysis was performed on patients with moderate or severe FMR, the use of MRA was not associated with a lower risk of composite adverse events (Figure 7). The subgroup analyses using studies with LVEF < 50% did not yield different outcomes, since the use of MRA did not significantly affect the risk of all-cause mortality risk and composite adverse events. The sensitivity analyses excluding studies that reported having patients with multivalvular pathology did not lead to different outcomes than the main analyses.

#### 3.3.4. Quality of Evidence and Publication Bias Assessment

The quality of evidence of the meta-analysis, as assessed by the GRADE tool, is summarized in Appendix A. Except for one forest plot (Figure 2B), the inconsistency of results across studies was not significant. Indirectness and imprecision were not serious for the majority of the presented forest plots (Appendix A). Overall, the findings of these analyses were considered important with a level of certainty ranging from very low to moderate (Appendix A). The generated funnel plots are presented in Appendix A.

## 4. Discussion

The present meta-analysis provided information on the prognostic impact of different GDMT-HF categories in FMR by recruiting a total of 6715 patients with FMR from 12 studies. The main finding was that FMR patients receiving RASi had a 48% and 45% lower risk of all-cause mortality and composite outcome, respectively, whereas patients receiving BB had a 38% lower risk of all-cause mortality and the composite outcome in unadjusted pooled random effect models. Both RASi and BB retained their association with the composite outcome in pooled adjusted models. The prognostic benefit of using RASi or BB was also retained in the subgroup analysis including patients with moderate of severe FMR. Contrastingly, the use of MRA did not demonstrate a significant association with improved outcomes in patients with FMR (Figure 8).

The triad of RASi, BB, and MRA serve as the foundation pharmacotherapy of HF, particularly for patients with reduced left ventricular ejection fraction. Sodium-glucose co-transporter 2 inhibitors have recently been added to this therapy throughout the spectrum of ejection fraction creating the four medical pillars of GDMT-HF [32]. However, there are no available published data for the impact of any of the sodium-glucose co-transporter 2 inhibitors on the outcome of patients with FMR included in this meta-analysis. Nevertheless, the multicenter, double-blind, randomized EFFORT trial recently demonstrated that the sodium-glucose co-transporter 2 inhibitor ertugliflozin induced significant reductions in the severity of the FMR grade, while improving left ventricular global longitudinal strain and left atrial size in HF patients with mildly reduced ejection fraction [33]. These findings hold promise that the use of sodium-glucose co-transporter 2 inhibitors could favorably impact the prognosis of FMR patients.

### 4.1. Renin–Angiotensin System Inhibitors

The use of RASi in HF is well-known for mitigating left ventricular (LV) remodeling, which translates into an improved prognosis. Wong et al. studied the effect of valsartan on cardiac structure and function in a large cohort of 5,010 HF patients, and they demonstrated a significant decrease in left ventricular end-diastolic diameter and increase in LVEF with the use of valsartan [34].

There is ample evidence that the angiotensin-receptor/neprilysin inhibitor is superior to angiotensin-converting enzyme inhibitors and angiotensin receptor blockers in achieving cardiac reverse remodeling in patients with HF and reduced LVEF [35], leading to a reduction in FMR grade [11]. Kang et al. demonstrated in a randomized double-blind trial that sacubitril/valsartan induced a greater decrease in effective orifice area and regurgitant volume compared to valsartan in HF patients with FMR [11]. This meta-analysis expands the knowledge beyond the aforementioned literature by highlighting the independent association of RASi with a lower risk of the composite adverse outcome in FMR.

### 4.2. Beta-Blockers

BB are known to exert a favorable effect in patients with HF through their negative chronotropic, antiarrhythmic, and inotropic properties. In a randomized study of patients with LVEF < 40%, metroprolol induced significant reductions in left ventricular end-systolic volume and improved systolic function compared to the placebo [36]. This reverse remodeling can potentially diminish FMR by reducing the tethering forces and the annular dilation. Capomolla et al. demonstrated that the use of carvedilol was linked with significant reductions in the effective regurgitant orifice area and regurgitate volume, accompanied by improvements in systolic function and ventricular size in a cohort with chronic HF [37]. Similar findings were reported by Colet et al., where carvedilol reduced the FMR grade in up to 80% of HF patients with systolic dysfunction [38]. The effect on FMR can be visible within the first few months after introducing the BB [39]. The current meta-analysis adds to the literature, showing that the use of BB can improve prognosis and reduce the risk of adverse cardiovascular outcomes in FMR patients.

### 4.3. Mineralocorticoid Receptor Antagonists

The association of MRA with improved prognosis and reverse remodeling is well-established in patients with HF [40,41]. Beyond diuresis and afterload reduction, MRA are known to have an antifibrotic profile in HF. In a cohort of patients with congestive HF after acute myocardial infarction, eplerenone was shown to suppress myocardial collagen turnover changes, as suggested by significantly lower levels of amino-terminal propeptide of type I and type III procollagen in the eplerenone group [42]. In line with those findings, in a randomized, placebo-controlled trial Udelson et al. demonstrated that procollagen type I N-terminal propeptide was significantly reduced for patients receiving eplerenone [43]. However, the impact of MRA in the pathophysiology of FMR has not been comprehensively studied so far, while this meta-analysis failed to demonstrate a link between the use of MRA and better prognosis in patients with FMR.

### 4.4. GDMT Across the Whole LVEF Spectrum of HF

The current meta-analysis included HF patients with FMR throughout the whole spectrum of LVEF. Performing a subgroup analysis of the studies with HFmrEF and HFrEF patients did not show any differentiation from the main outcomes of this meta-analysis. However, the GDMT-HF approach aims to succeed in reverse LV remodeling and functional LV recovery in HFrEF, while in HFpEF aims to reduce LA pressure and prevent LA remodeling and fibrosis [40]. Hence, the GDMT-HF depends on the LVEF category. In the case of HFrEF and HFmrEF, it is based on four pillars out of which the BB and the RASi category are proven to have a positive impact on the outcome of patients with FMR according to this meta-analysis. In the case of HFpEF, the sodium-glucose co-transporter 2 inhibitors are indicated but there is a lack of data on their impact on patient outcome when FMR is present [44]. If the GDMT-HF fails, cardiac resynchronization therapy should be appraised in HFrEF, and transcatheter or surgical mitral valve repair/replacement may be considered for all the LVEF groups [45].

### 4.5. Cardiovascular Comorbidities and FMR in HF

Concomitant conditions such as atrial fibrillation (AF), coronary artery disease, arterial hypertension and pulmonary hypertension can affect the occurrence and progression of FMR. AF plays a significant role in atrial FMR by contributing to LA dilation, mitral annular enlargement, atrial myopathy, and impaired atrial contractility [45,46]. Arterial hypertension and coronary artery disease may cause LV remodeling and dysfunction, papillary muscle displacement, and leaflet tethering, leading to ventricular FMR [47,48]. Additionally, pulmonary hypertension increases RV afterload, which in turn, through ventricular interdependence, affects LV function and mitral valve dynamics [49]. Cardiac rhythm restoration in AF, revascularization in coronary artery disease, and pulmonary vasodilators for selected cases of pulmonary hypertension may reduce FMR and mitigate its evolution beyond GDMT-HF.

### 4.6. Clinical Implications

Although the treatment of mitral regurgitation is the surgical or transcatheter repair of the valve, in the case of FMR, the patient may be at extremely high surgical risk and the left ventricle may be severely dilated with profoundly reduced systolic function that renders the interventional treatment futile [8,50,51]. The results of the randomized trials COAPT and MITRA-FR support the highly debated postulation of intervening in the proportionate FMR [50,51]. When the left ventricle is severely remodeled (>220 mL or 120 mL/m^2^) or the FMR is a bystander of the pathology, as expressed by the ratio of regurgitant volume/left ventricular end-diastolic volume <20%, the valvular intervention is deemed ineffective in modifying the patient’s outcome [52,53]. While a lot of light has been shed on the role of intervention in FMR, there is a complete lack of randomized data on the prognostic role of GDMT-HF in FMR. Observational data have previously disclosed that medical therapy can treat up to 40% of severe FMR cases, which is linked with favorable prognosis [54]. This meta-analysis highlights for the first time that RASi and BB could be used in FMR to lower the risk for adverse events, regardless of FMR severity.

### 4.7. Limitations

There are several limitations of the present meta-analysis that should be acknowledged. Firstly, the robustness of its results is inherently limited by the inclusion of observational studies which were not primarily designed to investigate the association of pharmacotherapy with outcomes in FMR. However, this is due to the complete lack of randomized controlled trials, which highlights the aim and adds to the necessity of such an analysis.

To minimize the risk of residual confounding, meta-analysis of adjusted HRs was employed. However, forest plots investigating the use of MRA were limited by the pooling of unadjusted HRs, and, therefore, their results should be cautiously interpreted. Meta-regression could potentially help minimize the risk for residual confounding, however, the limited number of studies did not allow its implementation.

The independent role of the angiotensin-receptor neprilysin inhibitor or angiotensin-converting enzyme inhibitors/angiotensin receptor blockers in FMR could not be explored due to the pooling of these categories together in the studies included. Furthermore, the impact of GDMT-HF in the HFpEF subgroup could not be elucidated in the present meta-analysis due to the lack of appropriate data for sub-analysis.

## 5. Conclusions

The present meta-analysis suggests that the use of RASi and BB is associated with a lower risk of adverse events in patients with FMR. This positive effect was retained in the subgroup analysis of moderate and severe FMR. On the contrary, the use of MRA failed to show an association with outcomes in FMR. Despite these promising findings in favor of RASi and BB, this meta-analysis emphasizes the need for further large prospective randomized clinical trials to determine the prognostic role of GDMT-HF in FMR—especially for the patients with HF with preserved ejection fraction, since these drugs are not first line treatment for them—and ultimately identify FMR responders to drugs, which could potentially delay the interventional valvular repair for the time point of achieved reverse remodeling, thus increasing the chances for a successful transcatheter repair.

## Figures and Tables

**Figure 1 diagnostics-15-00598-f001:**
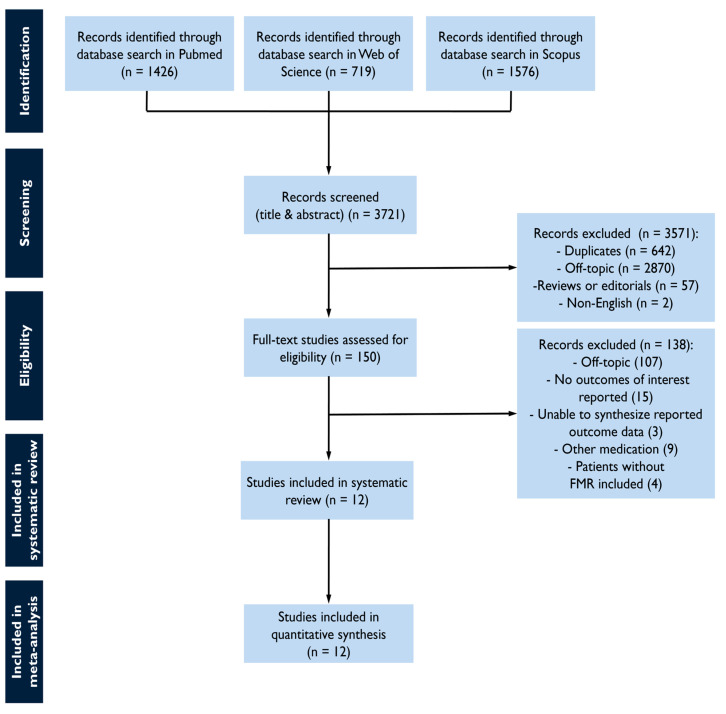
Study flow chart for study selection.

**Figure 2 diagnostics-15-00598-f002:**
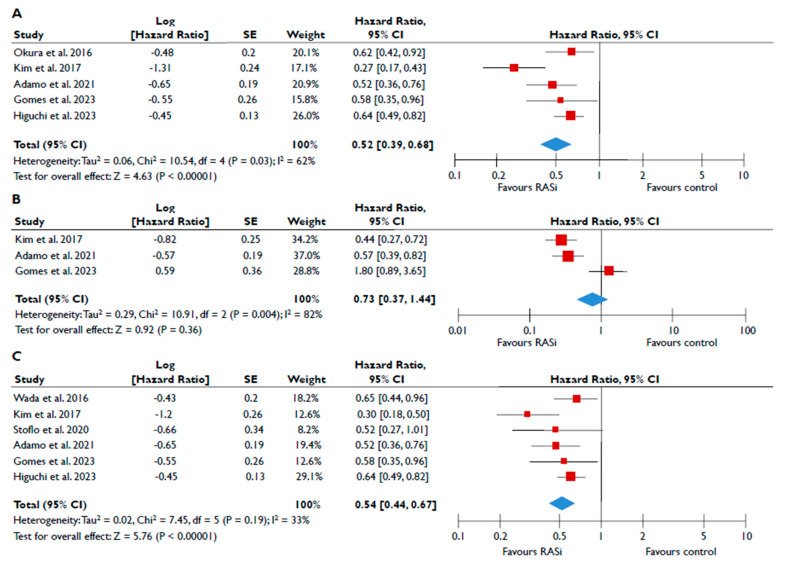
Association of RASi with the risk of events in patients with FMR. The forest plots display the pooled hazard ratio and the corresponding 95% confidence interval (CI) of each study, indicating the risk of all-cause death in unadjusted modes (**A**) and adjusted models (**B**), and the risk of the composite outcome in unadjusted (**C**) and adjusted (**D**) models for patients receiving RASi compared to patients not receiving RASi. Abbreviations: FMR, functional mitral regurgitation; RASi, renin–angiotensin system inhibitors [20,21,23,24,25,26,27,28,29].

**Figure 3 diagnostics-15-00598-f003:**
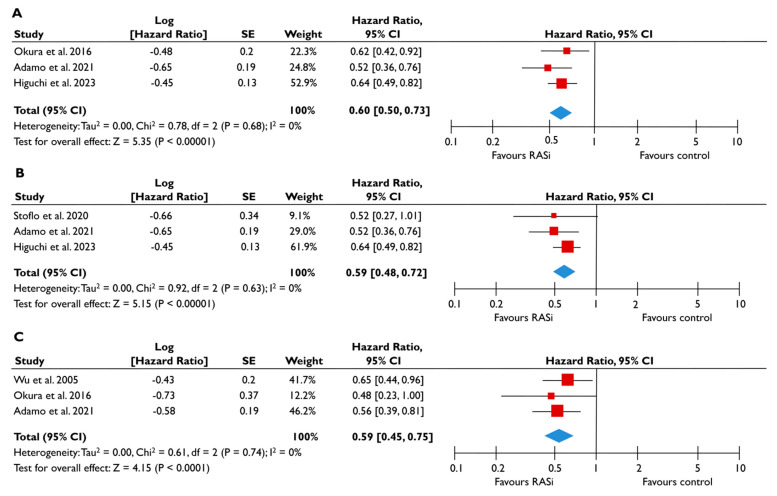
Association of RASi with the risk of events in patients with moderate or severe FMR. The forest plots display the pooled hazard ratio and the corresponding 95% confidence interval (CI) of each study, indicating the risk of all-cause death in unadjusted modes (**A**), and the risk of the composite outcome in unadjusted (**B**) and adjusted (**C**) models for patients receiving RASi compared to patients not receiving RASi. Abbreviations: FMR, functional mitral regurgitation; RASi, renin–angiotensin system inhibitors [21,23,24,28,29].

**Figure 4 diagnostics-15-00598-f004:**
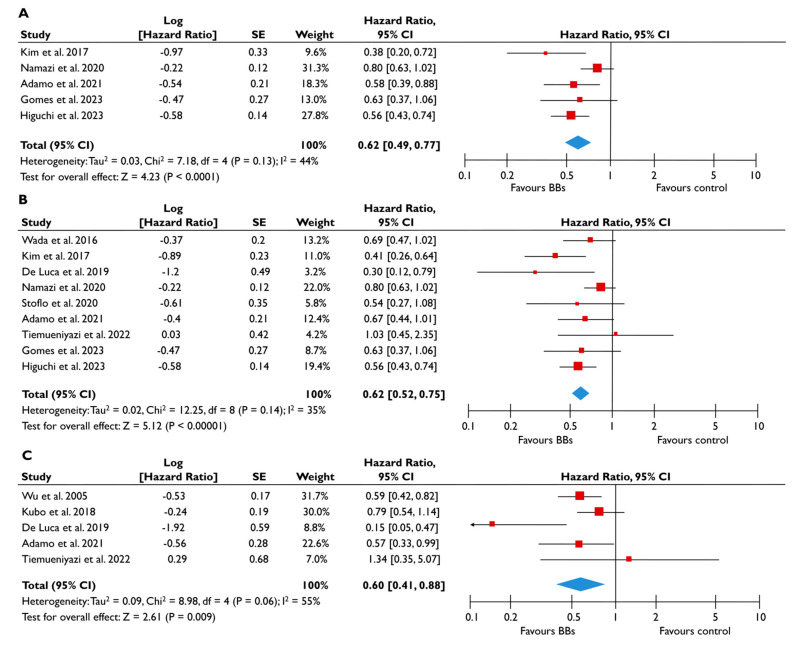
Association of BB with the risk of events in patients with FMR. The forest plots display the pooled hazard ratio and the corresponding 95% confidence interval (CI) of each study, indicating the risk of all-cause death in unadjusted modes (**A**), and the risk of the composite outcome in unadjusted (**B**) and adjusted (**C**) models for patients receiving BB compared to patients not receiving BB. Abbreviations: BB, beta-blockers; FMR, functional mitral regurgitation [20,22,23,24,25,26,27,28,29,30,31].

**Figure 5 diagnostics-15-00598-f005:**
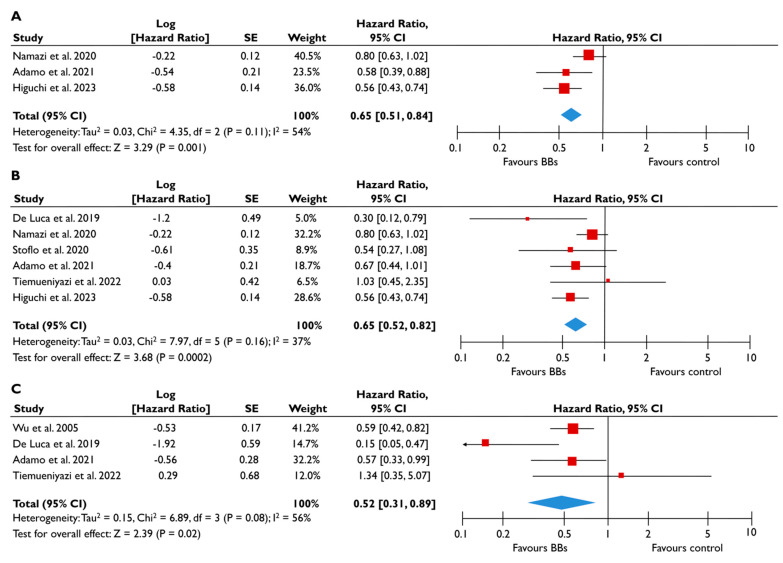
Association of BB with the risk of events in patients with moderate or severe FMR. The forest plots display the pooled hazard ratio (HR) and the corresponding 95% confidence interval (CI) of each study, indicating the risk for all-cause death in unadjusted modes (**A**), and the risk for the composite outcome in unadjusted (**B**) and adjusted (**C**) models for patients receiving BB compared to patients not receiving BB. Abbreviations: BB, beta-blockers; FMR, functional mitral regurgitation [22,23,24,28,29,30,31].

**Figure 6 diagnostics-15-00598-f006:**
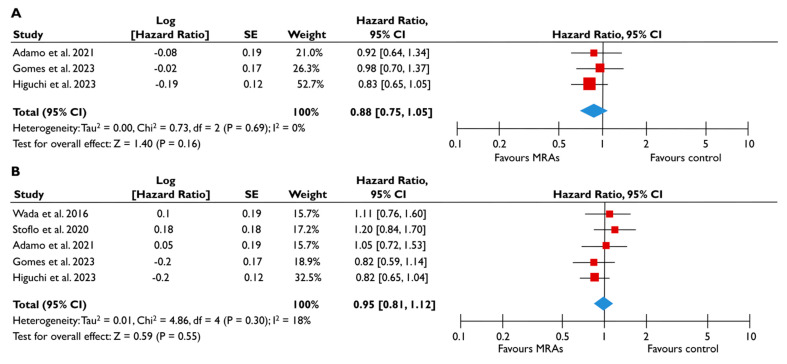
Association of MRA with the risk of events in patients with FMR. The forest plots display the pooled hazard ratio and the corresponding 95% confidence interval (CI) of each study, indicating the risk for all-cause death (**A**) and the risk for the composite outcome (**B**) in unadjusted modes for patients receiving MRA compared to patients not receiving MRA. Abbreviations: FMR, functional mitral regurgitation; MRA, mineralocortcoid receptor antagonists [23,24,25,27,28].

**Figure 7 diagnostics-15-00598-f007:**
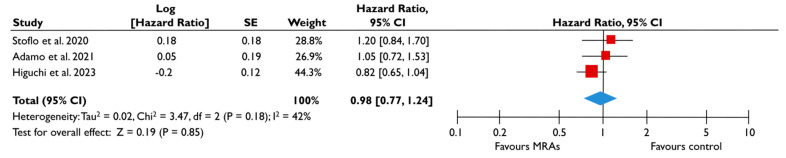
Association of MRA with the risk of events in patients with moderate or severe FMR. The forest plots display the pooled hazard ratio and the corresponding 95% confidence interval (CI) of each study, indicating the risk for the composite outcome in unadjusted modes for patients receiving MRA compared to patients not receiving MRA. Abbreviations: FMR, functional mitral regurgitation; MRA, mineralocortcoid receptor antagonists [23,24,28].

**Figure 8 diagnostics-15-00598-f008:**
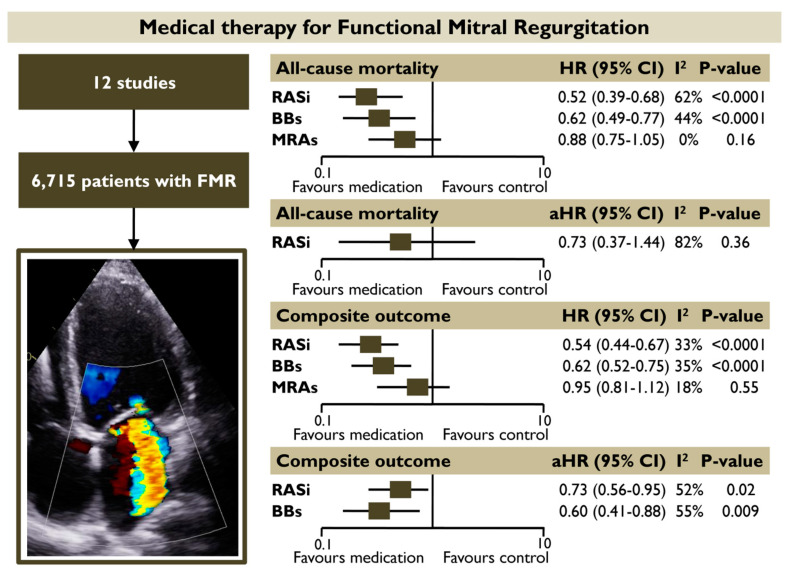
Prognostic impact of different medical categories for patients with FMR. The use of RASi and BBs was associated with a significantly lower risk of all-cause mortality and the composite outcome in pooled unadjusted models. Both RASi and BBs retained their association with the composite outcome in pooled adjusted models. MRAs did not demonstrate a significant association with improved outcomes. Abbreviations: BBs, beta-blockers; FMR, functional mitral regurgitation; HR, hazard ratio; RASi, renin-angiotensin system inhibitors; MRAs, mineralocorticoid receptor antagonists.

**Table 1 diagnostics-15-00598-t001:** Baseline characteristics of the included studies.

Author	Year	Population	Design	No. of Patients	Age, Years	Male, %	LVEF Cut-Off for Inclusion, %	LVEF, %	Medication	Follow-Up Period, Months
Wu et al. [29]	2005	At least moderate FMR	Retrospective, observational	682	63.5 ± 14.2	59.1	30	20.1 ± 7.0	RASi, BB	NS
Okura et al. [21]	2016	At least moderate ischemic FMR	Retrospective, observational	296	72.5 ± 11.6	60.1	NS	48.3 ± 11.2	RASi	NS
Wada et al. [25]	2016	Acute decompensated HF with FMR	Retrospective, observational	349	72.0 ± 13.0	65.3	NS	NS	RASi, BB, MRA	25.2 ± 15.6
Kim et al. [20]	2017	At least mild ischemic FMR	Retrospective, observational	551	68.1 ± 10.7	76.6	NS	50.4 ± 10.6	RASi, BB	61.2 (28.8, 86.4)
Kubo et al. [26]	2018	Acute decompensated HF with at least mild FMR	Prospective, observational	563	80.9 (72.3, 87.0)	54.5	NS	39.6 ± 18.9	RASi, BB	13.3 (7.7, 17.5)
De Luca et al. [31]	2019	Symptomatic HF with at least moderate FMR, no ischemia	Prospective, observational	33	71 (62, 76)	76.0	40	29.0 (26.0–36.0)	BB	25.0 (10.0, 40.0)
Stolfo et al. [28]	2020	Significant FMR undergoing TEER	Prospective, observational	1221	67.0 ± 12.0	77.0	40	30.8 ± 7.3	RASi, BB, MRA	25.0 (9.5, 40.5)
Namazi et al. [30]	2020	At least moderate FMR	Retrospective, observational	650	66.0 ± 11.0	68.0	NS	29.0 ± 10.0	BB	56.0 (28.0, 106.0)
Adamo et al. [23]	2021	Significant FMR undergoing TEER	Retrospective, observational	304	71.6 ± 9.4	74.0	50	32.2 ± 8.4	RASi, BB, MRA	21.5 (8.4, 36.8)
Tiemuerniyazi et al. [22]	2022	Moderate FMR and significant aortic valve disease	Retrospective, observational	165	59.2 ± 12.2	65.4	NS	55.0 (46.0, 61.0)	BB	18.4 (12.1, 18.3)
Higuchi et al. [24]	2023	Significant FMR undergoing TEER	Retrospective, observational	1594	74.0 ± 10.0	66.0	NS	35.0 ± 12.0	RASi, BB, MRA	20.4 (11.0, 37.1)
Gomes et al. [27]	2023	At least mild ventricular FMR	Retrospective, observational	307	70.0 (62.0, 77.0)	76.9	50	35 (27, 40)	RASi, BB, MRA	42.0 (16.8, 67.2)

Continuous variables are reported as median (interquartile range) or mean ± standard deviation and categorical variables as percentages. Abbreviations: BB, b-blockers; FMR, functional mitral regurgitation; HF, heart failure; MRA, mineralocorticoid receptor antagonists; NS, not specified; RASi, renin angiotensin system inhibitors; TEER, transcatheter edge to edge repair.

## Data Availability

The data underlying this article will be shared at reasonable request to the corresponding author.

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
