# Peer review of "Prognostic Implications of Guideline-Directed Medical Therapy for Heart Failure in Functional Mitral Regurgitation: A Systematic Review and Meta-Analysisâ€"

_diagnostics, 2025, doi:10.3390/diagnostics15050598_

Round 1
Reviewer 1 Report
Comments and Suggestions for Authors
The authors submitted a meta-analysis of the impact of conventional management on FMR in HF patients. They found that RASS inhibitors and beta-blockers improved FMR regardless of FMR severity. The findings are impressive and practically useful. However, I would like to make some comments to discuss them.
- In the section Introduction the authors might report prevalence of FMR depending o HF phenotypes and its relation to adverse clinical outcomes.
- Management of HF on FMR recovery / improvement is likety to be related to HF phenotypes and the authors are welcome to consider reportig the finidngs in connection with LVEF.
- The authors should discuss the role of concomitant conditions such as atrial fibrillation in FMR occurrence and the benefit of the management.
Author Response
Reviewer’s comments:
The authors submitted a meta-analysis of the impact of conventional management on FMR in HF patients. They found that RASS inhibitors and beta-blockers improved FMR regardless of FMR severity. The findings are impressive and practically useful.
Answer:
We thank the Reviewer for his/her comments, the positive feedback and detailed input on our study.
However, I would like to make some comments to discuss them.
- In the section Introduction the authors might report prevalence of FMR depending on HF phenotypes and its relation to adverse clinical outcomes.
Answer:
We thank the Reviewer for this comment. We have now added the following sentences in the revised Introduction section of the manuscript:
Lines 46-50:
“More specifically, the European Society of Cardiology (ESC) Heart Failure Long-Term Registry showed a prevalence of FMR up to 35% in the HF with reduced ejection fraction (HFrEF) group, 30% in the HF with mildly reduced EF (HFmrEF) group, and 20% in the HF with preserved EF (HFpEF) group (2). Other studies have also provided a prevalence of severe FMR up to 45% in patients with advanced HFrEF (3, 4).”
The relation of FMR to adverse clinical outcomes is also reported in the Introduction, lines 50-52:
“Any grade of FMR is independently associated with adverse outcomes in HF and confers a poor prognostic course throughout the range of left ventricular EF (LVEF)(5-7)”
- Management of HF on FMR recovery / improvement is likety to be related to HF phenotypes and the authors are welcome to consider reportig the finidngs in connection with LVEF.
Answer:
We thank the Reviewer for this constructive comment. We have conducted a subgroup analysis taking into consideration only studies with patients having LVEF <50%.
This is described in the revised Methods section:
Lines 135-138:
“…subgroup analyses were performed addressing the risk of all-cause mortality and/or the composite outcome only 1) for patients with significant (moderate or severe) FMR and 2) for patients with LVEF<50%. A sensitivity analysis excluding studies that reported to include patients with multivalvular pathology, was conducted.”
The outcomes of this subgroup analysis are described in the revised Results section as below:
Lines 193-196:
“In the subgroup analyses using only studies with LVEF <50%, the use of RASi was linked with decreased risk of all-cause mortality risk (in the unadjusted models), and with decreased risk of the composite adverse events (in both unadjusted and adjusted models).”
Lines 223-224:
“In the subgroup analyses using only studies with LVEF<50%, the use of BB was linked with decreased risk of all-cause mortality risk and composite adverse events.”
Lines 248-250:
“The subgroup analyses using only studies with LVEF<50% did not yield different outcomes since the use of MRA did not affect significantly the risk of all-cause mortality risk and composite adverse events.”
We have also added a paragraph in the revised Discussion section summarizing the FMR management across the whole LVEF spectrum in HF:
Lines 331-341:
“4.4. GDMT across the whole LVEF spectrum of HF
The current meta-analysis included HF patients with FMR throughout the whole spectrum of LVEF; performing a subgroup analysis of the studies with HFmrEF and HFrEF patients did not show any differentiation from the main outcomes of this meta-analysis. However, the GDMT-HF approach aims to succeed in the reverse LV remodeling and functional LV recovery in HFrEF while in HFpEF aims in reducing LA pressure and preventing LA remodeling and fibrosis (40). Hence, the GDMT-HF depends on the LVEF category; in case of HFrEF and HFmrEF is based on four pillars out of which the BB and the RASi category are proven to have positive impact on the outcome of patients with FMR according to this meta-analysis, in case of HFpEF the sodium-glucose co-transporter 2 inhibitors are indicated but there is a lack of data yet on their impact on patients outcome when FMR is present (44). If the GDMT-HF fails, cardiac resynchronization therapy should be appraised in HFrEF and transcatheter or surgical mitral valve repair/replacement may be considered for all the LVEF groups (45).”
- The authors should discuss the role of concomitant conditions such as atrial fibrillation in FMR occurrence and the benefit of the management.
Answer:
We thank the Reviewer for this insightful comment adding to the quality of our Discussion. We have now added a paragraph in the Discussion section briefly describing the role of concomitant cardiovascular conditions in FMR occurrence and progression:
Lines 343-356:
“4.5. Cardiovascular comorbidities and FMR in HF
Concomitant conditions such as atrial fibrillation (AF), coronary artery disease, arterial hypertension and pulmonary hypertension can affect the occurrence and progression of FMR. AF plays a significant role in atrial FMR by contributing to LA dilation, mitral annular enlargement, atrial myopathy and impaired atrial contractility(45, 46). Arterial hypertension and coronary artery disease may cause LV remodeling and dysfunction, papillary muscle displacement and leaflet tethering leading to ventricular FMR(47, 48). Additionally, pulmonary hypertension increases RV afterload, which in turn through ventricular interdependence affects LV function and mitral valve dynamics (49). Cardiac rhythm restoration in AF, revascularization in coronary artery disease, and pulmonary vasodilators for selected cases of pulmonary hypertension, may reduce FMR and mitigate its evolution, beyond GDMT-HF.”
Reviewer 2 Report
Comments and Suggestions for Authors
I recommend including all studies till 12/2024. There were also studies with ARNI and SGLT2 inhibitors in patients with functional MR. Thus, extend your search to other studies and include all of them. Studies with multi-valvular pathologies, e.g. aortic valve pathology and MR, should be excluded.
Author Response
Reviewer’s comments:
I recommend including all studies till 12/2024. There were also studies with ARNI and SGLT2 inhibitors in patients with functional MR. Thus, extend your search to other studies and include all of them. Studies with multi-valvular pathologies, e.g. aortic valve pathology and MR, should be excluded.
Answer:
We thank the Reviewer for his/her comments, the positive feedback and detailed input on our study.
Firstly, as per Reviewer’s suggestion, we have updated our literature search up to February 2025 aiming to find studies with ARNI and SGLT2 inhibitors in patients with functional MR. This search could not identify any more studies eligible for our meta-analysis. For instance, this significant study of 2024 published in the Circulation (DOI: 10.1161/CIRCULATIONAHA.124.069144; Ertugliflozin for Functional Mitral Regurgitation Associated With Heart Failure: EFFORT Trial) did not provide time-to-event data for follow-up adverse events. Nevertheless, the findings of this trial are discussed in our manuscript, in lines 278-283:
“…the multicenter, double-blind, randomized EFFORT trial recently demonstrated that the sodium-glucose co-transporter 2 inhibitor ertugliflozin induced significant reductions in the severity of FMR grade while improving left ventricular global longitudinal strain and left atrial size in HF patients with mildly-reduced ejection fraction (28). These findings hold promise that the use of sodium-glucose co-transporter 2 inhibitors could favorably impact the prognosis of FMR patients.”
Secondly, as per Reviewer’s suggestion, we have now conducted sensitivity analyses excluding from our meta-analysis studies reporting to have patients with multivalvular pathology (Tiemuerniyazi et al., Gomes et al., Stolfo et al., Wu et al. and Namazi et al.).
Revised Supplementary Table S2:
Reason for exclusion in the sensitivity analysis (n = 5 studies reporting to include patients with multivalvular heart disease) |
References of excluded studies |
1. Timerniyazi et al, as it included patients with FMR and concomitant aortic stenosis. |
(32) |
2. Gomez et al. reported to include patients (18% of the total population) with comorbid ΤR at baseline |
(33) |
3. Stolfo et al. reported to include patients (41% of the total population) with comorbid ΤR at baseline |
(34) |
4. Wu et al. reported to include patients (47% of the total population) with comorbid ΤR at baseline |
(35) |
5. Namazi et al. reported to include patients (18% of the total population) with prior tricuspid valvuloplasty at baseline |
(36) |
This is described in the revised Methods section:
Lines 136-138:
“A sensitivity analysis excluding studies that reported to include patients with multivalvular pathology, was conducted.”
The results of the sensitivity analyses are provided in the revised manuscript as follows:
RASi meta-analysis: Lines 196-198:
“Finally, the sensitivity analyses omitting studies reporting to include patients with multivalvular pathology did not yield different outcomes than the main analyses.”
BB meta-analysis: Lines 224-226:
“The sensitivity analyses excluding studies reporting to have patients with multivalvular pathology did not yield different outcomes than the main analyses.”
MRA meta-analysis: Lines 250-252:
“The sensitivity analyses excluding studies reporting to have patients with multivalvular pathology did not lead to different outcomes than the main analyses.”
Reviewer 3 Report
Comments and Suggestions for Authors
The presented systematic review and meta-analysis aimed to investigate the independent long-term prognostic impact of different medical categories in patients with heart failure (HF) and functional mitral regurgitation (FMR). The relevance of the research is apparent due to the fact, that the combination of HF and mitral valve insufficiency is widespread. Besides, FMR creates a vicious circle for HF patients, and this group requires special attention by clinicians. The presented analysis is novel, since previously only the comparison of surgical mitral valve repair and medical therapy was performed (10.31083/j.rcm2502048.). The types of medical therapy which were analyzed in the article are currently recommended for patients with HF. However, their significance in patients with HF and FMR was not specifically studied. The rationale for the review is presented in introduction in details. The inclusion and exclusion criteria, keywords and databases for literature search were selected adequately to the study purpose. The figures and the tables are presented clearly. The article is well written and will be of interest to readers. I have no any comments.
Author Response
Reviewer’s comments:
The presented systematic review and meta-analysis aimed to investigate the independent long-term prognostic impact of different medical categories in patients with heart failure (HF) and functional mitral regurgitation (FMR). The relevance of the research is apparent due to the fact, that the combination of HF and mitral valve insufficiency is widespread. Besides, FMR creates a vicious circle for HF patients, and this group requires special attention by clinicians. The presented analysis is novel, since previously only the comparison of surgical mitral valve repair and medical therapy was performed (10.31083/j.rcm2502048.). The types of medical therapy which were analyzed in the article are currently recommended for patients with HF. However, their significance in patients with HF and FMR was not specifically studied. The rationale for the review is presented in introduction in details. The inclusion and exclusion criteria, keywords and databases for literature search were selected adequately to the study purpose. The figures and the tables are presented clearly. The article is well written and will be of interest to readers. I have no any comments.
Answer:
We thank the Reviewer for his/her insightful appraisal and the positive feedback.
Round 2
Reviewer 2 Report
Comments and Suggestions for Authors
The authors reasonably replied to all my previous comments. The paper has significantly improved, so I don't have any more comments.